# Multidrug-Resistant *Salmonella* Species and Their Mobile Genetic Elements from Poultry Farm Environments in Malaysia

**DOI:** 10.3390/antibiotics12081330

**Published:** 2023-08-18

**Authors:** Syahidiah Syed Abu Thahir, Sakshaleni Rajendiran, Rafiza Shaharudin, Yuvaneswary Veloo

**Affiliations:** Environmental Health Research Centre, Institute for Medical Research, National Institute of Health, Ministry of Health, Setia Alam, Shah Alam 40107, Malaysiayuvanes@moh.gov.my (Y.V.)

**Keywords:** multidrug-resistant *Salmonella*, antimicrobial resistance, poultry, environmental microbiology, beta-lactams, quinolone resistant-determining regions (QRDR)

## Abstract

The prevalence and persistent outbreaks of multidrug-resistant (MDR) *Salmonella* in low-income countries have received growing attention among the public and scientific community. Notably, the excessive use of antibiotics in chicken feed for the purpose of treatment or as prophylaxis in the poultry industry have led to a rising rate of antimicrobial resistance. Therefore, this study aimed to determine the presence of antimicrobial-resistant *Salmonella* species and its mobile genetic elements from soil and effluent samples of 33 randomly selected poultry farms in Selangor, Malaysia. *Salmonella* species were isolated on selective media (CHROMagar™ *Salmonella*). VITEK^®^ 2 system was used to identify the isolates and their antimicrobial susceptibility. Subsequently, eight isolates were subjected to the whole genome sequencing (WGS). Based on the results, *Salmonella* spp. was detected in 38.1% (24/63) of samples, with the highest resistance to ampicillin (62.5%), followed by ampicillin/sulbactam (50.0%) and ciprofloxacin (45.8%). Meanwhile, the identified serovars were *Salmonella enterica* subspecies *enterica* serovar Weltevreden (*S.* Weltevreden), *S*. Jedburgh, and *S.* Brancaster. The most prevalent resistance genes detected include *qnrS1*, *bla*_TEM-176_, *dfrA14*, and *tet*(A). The *IncX*1 plasmid, with encoded resistance genes, was also detected in four isolates. Furthermore, mutations in the quinolone resistant-determining regions (QRDR) were discovered, specifically in the *gyrA, gyrB*, and *parC* genes. In short, surveillance such as continuous monitoring of antimicrobial resistance and emerging trends in resistance patterns through farm environmental samples could provide information to formulate public health interventions for effective infection prevention and disease control.

## 1. Introduction

*Salmonella* is a Gram-negative *Enterobacteriaceae* that causes various infections, mainly salmonellosis. *Salmonella enterica* serovar Typhi (*S*. Typhi) is of human origin and causes typhoid fever [1], while non-typhoidal *Salmonella* (NTS) serovar causes salmonellosis and is frequently zoonotic [2,3]. As such, global NTS infection is estimated to be up to 550 million cases and an annual death toll of 77,000 people [4]. *Salmonella* infection may originate from humans or animals, as well as food sources such as fruits, vegetables, meat, poultry products, and raw or undercooked eggs [5]. Improper food handling and unhygienic food practises could lead to a higher risk of salmonellosis with acute symptoms related to the gastrointestinal tract (nausea, vomiting, and diarrhoea), high fever, and abdominal cramps [6]. Its recovery is self-limiting in the absence of specific treatments or antibiotics. However, antibiotic treatment is required, particularly for children, older people, and immunosuppressed patients, for resistant organisms and invasive diseases [7,8]. 

To date, over 2500 host-specific *Salmonella* serovars have been identified. For example, serovars *S.* Dublin and *S.* Typhimurium are found in cattle, *S.* Enteritidis and *S.* Gallinarum in poultry, and *S.* Choleraesuis in swine. Concurrently, the emergence of multidrug-resistant (MDR) *Salmonella* in poultry farms has been reported around the world [9], including Malaysia [10]. The use of antimicrobials as growth promoters in chicken feed has been prohibited in the United States [11] and China [12], and effective intervention with policies helps reduce the use of antimicrobials in livestock. Due to laudable efforts and actions taken to prevent misuse, the use of antimicrobials has decreased by 43% in the European Union (EU) [13]. Despite the efforts, it is estimated that 99,502 tonnes of antimicrobials were used globally in 2020, and that number will rise to 107,472 tonnes by 2030, with Asian countries accounting for the majority of users [14]. With a combined 58% global contribution, the top 5 consumers of antimicrobials in 2020 were China, Brazil, India, the USA, and Australia [14]. Therefore, misuse of antimicrobials in the poultry industry with an extensive reservoir of bacteria would stimulate greater antimicrobial resistance (AMR) through selective pressures [15]. 

Remarkably, bacteria can acquire resistance genes and virulence factors from host cells through the transfer of mobile genetic elements (MGEs), such as plasmids, transposons, and phages, via transduction, conjugation, and transformation processes [16]. Since MGEs can cross species boundaries, the pool of MDR bacteria in poultry environments provides an excellent platform for bacteria to transfer critical MGEs for their survival [17]. For instance, consuming contaminated raw chicken or egg products leads to the transmission of MDR *Salmonella* to humans [9]. Similarly, faecal-oral transmission to humans occurs due to the discharge of poultry effluent into rivers or poultry droppings as fertiliser in vegetable irrigation, resulting in farm-to-fork transmission [18]. 

NTS cases are insufficiently documented in Malaysia, with most reported cases focusing on typhoidal illnesses. A recent study in Borneo revealed the presence of invasive NTS, where the annual incidence was 32.4 per 100,000 children under the age of five [19]. A retrospective investigation in the paediatric wards of Selayang Hospital also discovered that NTS contributed to 16% of childhood bacteraemia cases compared to 2.3% of *S*. Typhi-caused bacteraemia [20]. Besides, the Malaysian National Surveillance on Antimicrobial Resistance (NSAR) 2021 report on the resistance rate of *Salmonella* spp. from blood samples from 2020 to 2021 stated an increase in resistance to ampicillin (from 14% to 16.8%), co-trimoxazole (4.1% to 5.7%), chloramphenicol (4.1% to 5.7%), ceftriaxone (0.8% to 1.1%), and ciprofloxacin (0.8% to 1.0%) [21]. Apart from that, a study on poultry meat in Malaysia demonstrated up to 40% *Salmonella* prevalence in supermarkets, wet markets, or butcheries, with *S*. Enteritidis being the prominent serovar [22]. In another local study, 8.7% of NTS was recovered from retail markets and vegetable farms, with the highest resistance to ampicillin (20.7%), co-trimoxazole (17.2%), and chloramphenicol (15.5%). The study also recorded resistance to colistin and ertapenem [23].

With the advancement of bioinformatics analysis, various methods can be used for taxonomic identification, antimicrobial gene analysis, and MGE identification. Next-generation sequencing (NGS) is one of the leading technologies that could be efficiently utilised for diagnostics and epidemiological investigations given its outstanding performance, such as short turn-over time, cost-effectiveness, and the ability for whole genomic sequencing (WGS) or targeted DNA identification [24,25]. 

Previously published studies in Malaysia focused mainly on identifying *Salmonella* spp. in poultry meat or cloacal swabs. Realising the need to gain further understanding of the potential threat of *Salmonella* in poultry environments, this study aims to explore the presence of antimicrobial-resistant *Salmonella* serovars from soil and effluent samples of poultry farms in Selangor, Malaysia, and assess their mobile genetic elements (MGEs). Briefly, the detection rate of *Salmonella* spp. in soil and effluent from poultry environments was determined by culture and sensitivity testing. Subsequently, WGS was performed using selected isolates to identify the resistance profile, serovars, and MGE of *Salmonella* spp. 

## 2. Results

### 2.1. Antimicrobial Resistance

A total of 63 samples, comprising 33 soil samples and 30 effluent samples were collected from the poultry farms. The detection rate of *Salmonella* spp. was 38.1% (24/63), of which 45% (15/33) were isolated from soil samples and 30% (9/30) from effluent samples. However, there was no significant difference between soil and effluent samples (*p* value = 0.153). Thus, the results were discussed as environmental samples from poultry farms. Table 1 shows in detail the phenotype resistance of *Salmonella* spp. isolates. 

An organism is labelled MDR if it exhibits resistance to at least one agent in three or more groups of antibiotics [26]. In addition, the Multiple Antibiotic Resistance (MAR) index is one of the indicators used to analyse antibiotic resistance. A low-risk category is represented by a MAR index of less than 0.2, while a high-risk category is represented by a MAR index of more than 0.2 [27,28]. Based on these conventions, 66.7% (16/24) of the isolated *Salmonella* spp. in this study demonstrated an MDR pattern. The mean MAR index calculated was 0.42 (SD = 0.096), suggesting a high-risk application and contamination of antibiotics in the poultry farm. Table 2 lists the resistotypes of *Salmonella* spp.

### 2.2. Whole-Genome Sequencing (WGS)

Eight *Salmonella* spp. were subjected to WGS using the Miseq platform. The GC content for *Salmonella* spp. isolates ranged from 51.87% to 52.15%, with a mean of 52.06%, while the mean genome length was 48,944,962 bp. The mean N50 contig length was 338,493 (minimum 137,441 and maximum 456,144). There were an average of 4918 protein-coding genes (CDS). Furthermore, the GC content and genome length were comparable to those of *S.* Enteritidis, *S*. Typhimurium, *S*. Brancaster, and *S*. Weltevreden strains (Genome assembly ASM950v1, ASM694v2, ASM429172v1, ASM300011v1).

### 2.3. Sequence Types (STs) and Serovar Prediction 

The prediction of sequence types through PubMLST detected three strains belongs to ST 365 and five other strains belonging to ST 2133. Meanwhile, the serovar prediction was performed in silico using SeqSero2 and *Salmonella* In Silico Typing Resource (SISTR) [29,30] according to Kauffman and White’s scheme [31]. The SeqSero2 and SISTR predictions show an accuracy of 75% (n = 6). Comparatively, SISTR predicted two isolates, as *S.* Jedburgh or *S.* Llandoff, while SeqSero2 identified these two isolates as *S.* Jedburgh. All the other serovars were predicted similarly by Seqsero2 and SISTR. Only serovars predicted by Seqsero2 are shown in Table 3.

### 2.4. Antibiotics Resistance Genes (ARGs)

Of the 18 drugs tested on the antibiotic susceptibility testing (AST) card, the WGS analysis detected 15 corresponding genes using the Resfinder and Antibiotic Resistant Gene-Annotation (ARG-ANNOT) [32,33]. The *aph(3′)-Ia* was observed in 50% (n = 4) of the isolates (Table 4). Furthermore, one of the isolates (MYS11) has additional resistance genes conferring resistance to the aminoglycoside group, which consists of *aac(3)-IV* (gentamicin), *aph(4)-Ia* (hygromycin), and *ant(3″)-Ia* (streptomycin). The *bla*_TEM-176_ gene was detected in four isolates (50%) and showed resistance to ampicillin, while *bla*_TEM-1_ was detected in one isolate (12.5%). A study characterising the *bla*_TEM-176_ gene isolated from *Escherichia coli* D7111 (NG_050215.1) showed that it has a single mutation in A222V (alanine to valine) from the *bla*_TEM-1_ gene and phenotypical resistance to ampicillin [34]. Additionally, the ARG-ANNOT database identified two additional putative beta-lactam genes, *bla*_Penicillin Binding Protein *E. coli*_ and *ampC*-encoding *bla*_AmpH_, in all eight isolates (100%). In the presence of the *ampC* gene, exposure to beta-lactam antibiotics may lead to inducible resistance to beta-lactams [35,36].

Furthermore, four isolates (50%) and a single isolate (12.5%) encode genes for trimethoprim (*dfrA14*) and sulfamethoxazole (*sul3*), respectively, and phenotypically, all isolates were susceptible to these drugs. The other two antibiotic groups (tetracycline and chloramphenicol) were not tested phenotypically, but *tet*(A) and *floR* genes were identified in five isolates (62.5%). Additional *mph*(A) (macrolides), *lnu*(F) (lincomycin), and *fosA* (fosfomycin) genes were also detected in one isolate (12.5%).

Moreover, in the fluoroquinolone group, the *qnrS1* gene was detected in five isolates (62.5%), which were also phenotypically resistant to ciprofloxacin.

### 2.5. Chromosomal Point Mutation

At least one chromosomal mutation was detected in all isolates, specifically in QRDR, *parC*, which have a significant role in DNA replication (Table 4). Regarding the findings, this is the first study to report mutations in *gyrA* at codon G438A, *gyrB* at codon A295G, and *parC* at codon A395S. Therefore, the significance of resistance is unknown and could not be predicted from the observed phenotype due to insufficient evidence.

### 2.6. Plasmid Multi-Locus Sequence Typing (pMLST)

Based on the Plasminfinder and pMLST analyses, all isolates contain at least one plasmid, such as *IncX*1 (n = 4), *IncFII* (S) (n = 3), *Col156* (n = 1), *Col440I*, and *ColRNAI* (n = 2) (Table 4).

MYS11 possessed five types of plasmids, comprising *IncFIA* (HI1), *IncHI1A*-ST 16, with novel loci in HCM1.259.2*, *IncHI1B*(*R27*), *IncN*-ST 3, 12, and *Col440I*. The *IncHI1A* plasmid from the MYS11 isolate carries a locus for common gene actions, such as HCM1.043, HCM1.064, HCM1.099, HCM1.116, and HCM1.259.2 (coded with similar alleles). The pMYS11 plasmid recorded a total length and GC content of 21,962 bp and 51.87%, respectively. Intriguingly, this plasmid is similar to those reported in the *Escherichia coli* strain from the chicken liver sample (NZ_CP016183.1), the pig faecal swab, which also encodes the *mcr*-1 gene isolated in Malaysia (NZ_CP016184.1), the *E. coli* strain isolated from the human faecal sample in Singapore (NZ_CP070903.1), *Salmonella* spp. isolated from the human faecal sample in China (NZ_CP060586.1), and *S*. Derby from the pork sample in Vietnam (NZ_CP068510.1) (refer to Appendix A). 

### 2.7. Mobile Genetic Elements (MGEs) and Salmonella Pathogenic Island (SPI)

Further analysis of *Salmonella* spp. detected the presence of plasmids encoding MGEs, with the incompatible group X plasmid (*IncX*1) (n = 4) encoded with the resistance genes *aph* (3′) *Ia* and *bla*_TEM-176_, with an additional two MGEs within the same contigs (Table 4). Other MGEs found within the same isolates include insertion sequences (IS) and transposons (Tn), which may be responsible for the transfer of resistance genes. Apart from that, IS102 and Tn 6024 were the most common transposons detected in *Salmonella* spp. (n = 4). Up to 10 SPIs were detected among the *Salmonella* spp. (Table 4). SPI-3 (100%), SPI-5, and SPI-9 (75% each) were the most predominant genes among the isolates. 

### 2.8. Integron

Integrons refer to specific ports for constructing external open reading frames via site-specific recombination, producing functional genes with the proper expression [37]. Based on the IntegronFinder, five strains (62.5%) harboured class 1 integrons (Int1). Table 4 shows that isolate MYS11 harbours two gene cassettes in two contigs with one resistance gene (*sul*3). In contrast, only four strains possessed the integron integrase (In0) elements, but no attC sites were detected (Appendix A). 

### 2.9. Phylogenetic Tree

A comparative analysis was performed to evaluate the distribution of *Salmonella* spp. serovar based on closely related *Salmonella* species deposited in the NCBI database (Pathogen Detection Isolates Browser). Note that the accession number is provided in the Appendix A. Subsequently, the single nucleotide polymorphism (SNP) matrix was generated, with a minimum and maximum of three and 32,289 SNPs, respectively, and each *Salmonella* spp. isolate in this study possessed a minimum of six different SNPs (refer to Appendix A). The phylogenetic tree was then constructed using reference genomes of *E. coli* (GCF_000008865.2), *S*. *bongori* (GCF_000439255.1), and *S.* Typhimurium (GCF_000006945.2) to root the tree, as illustrated in Figure 1. Additionally, a set of six genomes from *S. enterica* (GCF_016018515.1, GCF_016017915.1, GCF_016018525.1, GCF_016018455.1, GCF_016018765.1, and GCF_016017455.1) was used to outgroup the tree. 

The constructed tree contains two main nodes comprising monophyletic and paraphyletic groups. Group C forms a monophyletic group that includes *S.* Weltevreden isolates of ST 365 and is closely related to isolates from Thailand, Vietnam, China, and Malaysia. The SNP distance within these isolates ranges from 21 to 91 SNPs. Meanwhile, Group E forms the main subclade containing *S.* Brancaster isolated from this study and is closely related to strains from Singapore (environment), the United Kingdom (clinical), Taiwan (human), and Malaysia (human and chicken). The SNP’s distance within these isolates ranges from 6 to 53 (refer to Appendix A). 

The lowest SNP distance of 6 to 9 was detected between the MYS15 strain isolated from this study, three isolates of Taiwanese origin (DAAQFQ010000001.1, DAAQGX010000001.1, and DAAQMI010000001.1), and a single isolate from Singapore (QAUM01000001.1). In general, a SNP distance of less than ten indicates that the isolates are genetically similar, implying that they share a recent common ancestor or a common source of infection. Conversely, SNP distances greater than ten denote that the isolates have distant ancestors and potentially long environmental-food-human relations [38]. 

Although a complete genome for *S.* Jedburgh was unavailable in the NCBI database to compare and infer the phylogeny, the findings in this study demonstrate that two isolates of *S.* Jedburgh serovar (Group D) were highly distinct from any isolated serovar and remained in the node as Group E, possibly sharing a similar ancestor with *S*. Brancaster. Interestingly, *S.* Jedburgh has the same sequence type (ST 2133) as other *S.* Brancaster serovars.

## 3. Discussion

Efforts to detect *Salmonella* spp. in poultry environments have been extensively intensified given that the poultry industry is a major food resource for eggs and meat and a critical reservoir for *Salmonella* outbreaks. Previously, local researchers from the east coast of Peninsular Malaysia discovered a higher detection rate of *Salmonella* spp. in cloacal swabs (46.3%) and faecal samples (59.5%) compared to those in sewage samples (35.7%) and tap water samples (14.3%) from poultry environments [39]. These findings are consistent with the low detection rate of *Salmonella* spp. from effluent findings (30%) in the present study. Another study conducted in a wet poultry market found that all its environmental samples (floor, drain swab, drain water, display table, knife, and bench wash) were positive for *Salmonella* spp. [40]. This could probably be due to the cross-contamination of chicken meat in these environments. Comparatively, the detection rate of *Salmonella* spp. in the present study was 38.1%, whereas other local studies reported varying detection rates of 6.5–88.46% [40,41,42]. In other Asian countries, the detection rate of *Salmonella* spp. was 28.7%, 28.8%, 42.8%, and 48.7% for Thailand [43], Singapore [44], Cambodia [43], and Vietnam, respectively [45]. The varying results from these studies could be due to the different sampling locations (market or processing plant), sample types (meat, chicken cloacal swab, and environmental swab), and climate conditions influenced by geographic location, which may affect the overall prevalence rate of *Salmonella*. 

The most common NTS serovar is *S.* Enteritidis, which is abundantly found in foods, animals, and humans across various countries. The serovars of *S.* Brancaster and *S.* Weltevreden detected in the present study were similar to strains found in poultry farms by other Malaysian researchers [40,46] and in Asian countries, such as Singapore [47] and Vietnam [48]. Furthermore, *S.* Weltevreden was the second most common NTS serovar in Malaysia from 2003 to 2005 [49,50,51], and *S.* Weltevreden was the commonest serovar responsible for salmonellosis in Thailand from 1993 to 2002 [52]. 

AMR remains a major burden for the healthcare and food-animal production industries. Indigenous use of antibiotics leads to relentless cycles of AMR and exhausts the last resort of antimicrobials. In the present study, a high detection rate of resistance was observed among *Salmonella* spp. to ampicillin. Several Chinese states have reported similar resistance rates ranging from 57.9% to 98.4% [53,54,55], as well as Singapore at 78.8% [44] and Thailand at 72.4% [43]. However, a lower resistance rate was reported in Vietnamese poultry farms at 41.6–54.14% [45,56]. The high resistance to ampicillin may indicate its overuse or misuse in animal agriculture. Thus, enforcing proactive measures, such as the occasional use of antimicrobials when necessary, avoiding the addition of antibiotics for growth promotion, and implementing strict biosecurity measures to prevent the spread of diseases, are crucial to reducing the emergence of ampicillin-resistant *Salmonella* in poultry farms. 

Besides ampicillin, high resistance to ciprofloxacin (48%) was observed in this study. Contrary to the present findings, a recent study conducted on the east coast of Peninsular Malaysia revealed that 4.8% of the isolates were resistant to ciprofloxacin [42], while samples collected in the poultry processing wet markets in Penang and Perlis noted 3.5% resistance to ciprofloxacin [10]. Moreover, Abatcha et al. reported that all their isolates were sensitive to ciprofloxacin [41]. In other countries, such as Egypt and Ethiopia, the resistance rates were 30.8% and 29.3%, respectively [57,58]. However, the resistance rate was comparatively lower in China (12.5%) [55], possibly due to good food hygiene practises in retail meat, with effective surveillance systems in supermarkets and open-air markets. Ciprofloxacin and cephalosporin are frequently used to treat severe salmonellosis caused by NTS. Nevertheless, the current resistance trend is worrying, as the range of available drugs to prevent a severe infection will diminish in line with increasing resistance rates. On this basis, further studies involving a large sample size are required to ascertain the prevalence of fluoroquinolone resistance in Malaysia. 

Based on the present findings, 66% of the isolates were MDR *Salmonella*, while 82% were MDR *Salmonella* from broiler farms [42] reflecting a variety of MDR patterns in Malaysia. To prevent the spread of MDR bacteria to humans, appropriate authorities should oversee the distribution and application of prophylaxis to prevent unregulated and misuse of such antimicrobials in food chain production. Besides, improving animal husbandry practises, such as providing good nutrition and implementing disease control and prevention strategies, would effectively address the rising microbial resistance rate.

Furthermore, the resistance genes detected in this study were also found in poultry and environmental samples from wet markets in Penang and Perlis however, some additional genes were detected from the studies, such as beta -lactams (*bla*_PSE-1_, *bla*_TEM-B_), sulphonamide (*sul*1, *sul*2), tetracycline [*tet*(A), tet(B), *tet*(G)], aminoglycoside (*strA*, *aadA*), quinolones (*qnrA, qnrS*), and chloramphenicol genes (*floR*, *cmlA*) [10]. Another comparative study collected from human, poultry, and food samples recorded 10% resistance to beta-lactams (*bla*_TEM33_, *bla*_TEM4_, and *bla*_CTX-M_) and detected *dfrA14*, *dfrA15*, *sul*2, and *floR* genes in several isolates [59]. A study in China also discovered a high incidence of the *bla*_CTX-M_ genes in human and chicken meat [60]. Although resistance to colistin and the *mcr-1* gene had been previously reported in Bangladesh [61], none of the isolates in this present study possessed *bla*_CTX-M_ or the *mcr-1* gene. It is known that *Salmonella* is infectious if ingested with contaminated poultry products. The presence of various ARGs in a pool of poultry products can further impede human treatment when infected, resulting in the use of broad-spectrum antibiotics. The rate of mortality will also increase if treatment is delayed. Due to the small sample size, the coincident rate between the phenotype and genotype could not be calculated. Thus, this subject will be addressed in future studies.

Interestingly, *S.* Weltevreden from this study only carried the chromosomal cryptic gene *aac(6′)-Iaa*, which is found in *salmonella* and confers resistance to aminoglycoside. Likewise, genomic characterisation done with *S.* Weltevreden from human and non-human origins found that not many genes were detected from this serovar [62,63]. A few ARGs that were detected from non-human origins include *aac(6′)-Iaa*, *aac(6′)-Ib-cr*, *aph(3″)-Ib*, *aph(6)-Id*, *sul*1, *sul*2, *tet*(A), and *aac(6′)-Ib-cr* genes [64]. This indicates that *S*. Weltevreden did not display high AMR, suggesting that a wide range of antimicrobials are still available for treating the diseases caused by this serovar. 

Florfenicol is a synthetic drug with a fluorinated analogue of chloramphenicol (Cm), which is a plasmid-or chromosomal-encoded Cm exporter (efflux pump) usually found in *E. coli*, *Klebsiella pneumonia*, and *S.* Typhimurium. In this present study, 62.5% of the isolates were found to carry the *floR* gene, denoting resistance to chloramphenicol and florfenicol. Legally, nitrofuran, beta-agonist, and chloramphenicol drugs have been banned in veterinary and farming practices in Malaysia [65]. The tetracycline *tet*(A) gene was also predominantly detected in 62.5% of this study’s isolates, suggesting an extensive application of tetracycline in the poultry industry. Similarly, veterinary use of tetracycline has been banned in Malaysia since 2019 [66]. While these studies could be used as baseline data on resistance genes before the bans, further studies could be carried out to determine the AMR and ARG rates and observe their effectiveness. 

Mutations in DNA gyrase (*gyrA* and *gyrB*) and topoisomerase IV enzymes (*parC* and *parE*) can alter the active site of the QRDR and inhibit the binding of quinolones. For instance, the presence of a mutation in *gyrA*: E438A and *gyrB*: A295G with a double mutation in *parC*: N395S and T57S were detected in five of the isolates with MIC to ciprofloxacin of 0.5 to 1 μg/mL from this current study. While two isolates with a MIC of 0.5 μg/mL (intermediate susceptibility) have mutations in *gyrA*, *gyrB*, and *parC*, phenotypic testing shows that both have a reduced affinity for the drug. Similarly, a study from Singapore revealed mutations in *gyrA* (D87N, S83Y), which were associated with reduced affinity and resistance to ciprofloxacin and nalidixic acid [44]. As the simultaneous mutation in different codons was only reported in these studies, the relationships between these mutations and drug affinity must be further investigated.

A reduced quinolone uptake by *Salmonella* was also achieved via cell membrane alteration or overexpression of efflux pumps, such as energy-dependent efflux pumps, for example, the *AcrAB-TolC* efflux pump, although their role is limited, and the desired effect is achieved when the target enzyme undergoes simultaneous modification. In this study, mutations in the efflux pump *AcrB* occurred in all isolates, while only five isolates experienced simultaneous mutations involving *gyrA*, *gyrB*, and *parC*. The observed mutation was presumed to trigger significantly stronger quinolone resistance among *Salmonella*. Besides, a triple QRDR mutation in *gyrA* S83, D87, and *parC* S80I was reported in a study in Egypt, which led to considerable resistance to fluoroquinolones with an increased MIC value towards ciprofloxacin [67].

Aside from the presence of several other resistance genes, such as *aph(3′) Ia*, *bla*_TEM-176_, and transposons, four isolates (*S.* Brancaster and *S.* Jedburgh) were found to carry plasmid *IncX*1. This plasmid was first isolated in animals, suggesting that animals are a significant reservoir of the *IncX* plasmid. The *IncX* has also been frequently detected in *Salmonella* spp., which codes for quinolone, beta-lactam (*bla*_TEM-1_, *bla*_SHV-11_, and *bla*_CTX-M-1_), and carbapenem-resistance genes (*bla*_NDM_, *bla*_KPC_) [68,69,70]. However, an incompatible group of plasmids (*Inc*), known as the resistance factor (R factor), is capable of conjugating and transferring resistance DNA independently of the bacterial host and is significantly present in bacterial pools, such as the human gut [71]. The presence of the *Inc* plasmid in the isolates facilitates transmission within the poultry environment and through other hosts, including humans. The comparison of the *IncX*1 from this study with the PLSDB database recorded a 99.9% similarity to the *E. coli* plasmid p40EC-8 (NZ_CP070928.1) isolated from human samples and *S.* Brancaster isolated from chicken samples, both from Singapore (NZ_CP037995.1) (Appendix A). Thus, the result suggests a possible long interlinkage between the *Inc* plasmid and the isolates. 

*Salmonella* SPI is a set of virulence genes responsible for the pathogenicity and virulence of *Salmonella*. Located mostly within its chromosomes, the key part of SPI virulence is encoded from SPI-1 to SPI-5. In particular, SPI-1 and SPI-2 encode the Type-3 Secretion System (T3SS) responsible for invasion, proliferation in host cells [72]. SPI-3 is responsible for intramacrophage replication, while SPI-4 functions in epithelial adhesion and colonisation, and SPI-5 is involved in *Salmonella* enteropathogenicity [73]. Based on the results of the present study, *Salmonella* isolates could be considered pathogenic in poultry products and lead to severe infection in humans if ingested. 

Ultimately, the phylogenetic similarity of the isolated strains within these regions and other countries was analysed based on the SNP pairwise analysis. The constructed SNPs matrix revealed a casual association with SNP less than 50 branches into one node (Groups B, C, D, and E) with similar ST (*S.* Brancaster and *S.* Weltevreden). Moreover, the SNPs of *S.* Brancaster isolated in this study and those from Singapore, Taiwan, and the UK were found to be less than 10, and the phylogenetic tree converged into one group, suggesting a possible close ancestral origin. Despite a commendable outcome, the method applied in this study was more appropriate when examining epidemiological outbreaks of food-borne origin, hospital transmission outbreaks, or epidemiological investigations. 

## 4. Materials and Methods

### 4.1. Location of Sampling

Sites: Soil and effluent sampling (environmental) in this study was performed at thirty-three randomly selected poultry farms out of 212 registered with the Department of Veterinary Services (DVS), Selangor State, in 2017. Sample collection was carried out from January 2018 until October 2019. The inclusion criteria for the selection of poultry farms in this study included poultry farms that had been registered with the DVS, while the exclusion criteria were farms with mixed breeding of poultry and other livestock. 

Soil: Soil sub-samples were collected from three locations within the farm, particularly around the chicken cage. A metal spade was disinfected with 75% ethanol before collecting the soil sub-sample and transferring them to sterile plastic bags. Samples were kept in an ice box and transported to the laboratory, which was processed within 24 h of collection. The soil sub-sample was then homogenised into a single constitute at the laboratory [74]. In total, 33 pooled soil samples were taken for further analysis. 

Effluent: Effluent samples were collected from drainage pipes or, if available, puddle water around the poultry farm. A metal scoop was sterilised with ethanol and flamed on site, each time before collecting the samples and packed inside a sterile plastic bag. Samples were labelled, stored in an ice box, transported to the laboratory, and processed within 24 h of collection. A total of thirty pooled effluent samples were taken for analysis since no direct effluent source was available in three of the poultry farms. The effluent sample analysis was performed according to the standard method [75,76].

### 4.2. Enrichment and Isolation of Presumptive Salmonella

Enrichment and isolation of presumptive *Salmonella* followed the standard ISO 6579-1:2017 [77]. Primarily, 25 g of soil and effluent samples were enriched in 225 mL of Buffered Peptone Water (BPW) and incubated for 18 h under aerobic conditions at 37 °C. Then, 0.1 mL of the BPW culture was mixed with 10 mL of Rambaquick (RambaQUICK *Salmonella*, CHROMaga, Paris, France) and further incubated for at least 7 h under aerobic conditions at 41.5 °C. Then, 0.1 mL of broth was pipetted onto a *Salmonella* Plus plate (CHROMaga, Paris, France) and incubated for 18–24 h under aerobic conditions at 37 °C. The number of colony-forming units per mL (CFU/mL) in the chromagar plate was calculated, and a presumptive *Salmonella* isolate was selected from the plate, which appeared as a mauve colour and streaked on a Trypticase Soy Agar (TSA) plate to obtain a single-forming colony.

### 4.3. Identification and Susceptibility of Salmonella spp.

A single colony isolate from the TSA was first subjected to Gram staining for identification, followed by analysis using the VITEK^®^2 system; VITEX^®^2 GN ID cards (BioMerieux, Nurtingen, Germany) was used for the identification of *Salmonella*. Additionally, AST-GN83 cards were used according to the manufacturer’s instructions for antimicrobial susceptibility testing. Antibiotics tested on *Salmonella* included ampicillin (AMP), amoxicillin/clavulanic acid (AUG), ampicillin/sulbactam (SAM), piperacillin/tazobactam (PTZ), cefazolin (CF), cefuroxime (CXM), cefoxitin (FOX), cefotaxime (CTX), ceftazidime (CTZ), ceftriaxone (CEX), cefepime (CPM), aztreonam (AZT), meropenem (MPN), amikacin (AMI), gentamicin (GEN), ciprofloxacin (CIP), nitrofurantoin (NIT) and trimethoprim/sulfamethoxazole (SXT). The results were interpreted according to the Clinical and Laboratory Standards Institute (CLSI) guidelines.

### 4.4. DNA Extraction and Whole Genome Sequencing of Salmonella spp.

Approximately 1 mL of isolated *Salmonella* culture grown in brain heart infusion broth (incubated at 37 °C for 18–24 h) was taken for DNA extraction using the MasterPure Complete DNA and RNA Purification Kit (Lucigen, WI, USA) following the manufacturer’s instructions. The final extracted DNA was eluted with 35 μL nuclease-free water. The quantity and purity of DNA were further accessed with a Qubit 4 Fluorometers (Thermo Scientific, Waltham, MA, USA) and NanoDrop Spectrophotometers (Thermo Scientific, Waltham, MA, USA), followed by gel electrophoresis. Library preparation was performed using the genomic DNA technique with Illumina DNA PCR-Free Prep and DNA PCR-Free R1 Sequencing Primer (Illumina, San Diego, CA, USA) according to the manufacturer’s protocol. Sequencing was completed with the 500-cycle MiSeq Reagent Kit (v2) (Illumina, San Diego, CA, USA) with 100x coverage on the Miseq Illumina platform (Illumina, San Diego, CA, USA).

It is necessary to highlight certain limitations of the present study. In essence, only eight samples were subjected to WGS analysis. Apart from financial constraints, the relocation of the institute from late 2019 to 2020, during which the study was carried out, led to the loss of some samples gathered from 2018 to 2019 due to transportation errors and samples not maintained at optimal temperature during the transfer. Nevertheless, after obtaining sufficient DNA and financial support, the sequencing of the eight samples was performed in December 2021. 

### 4.5. Bioinformatics Analysis

Various bioinformatics software was utilised for the bioinformatics analysis. Initially, raw data were trimmed using Trimmomatic (version 0.38) [78] to ensure that the available data could be assembled using Velvet (version 1.2.10) [79], while PlasmidSpades (version 3.15.5) [80] was used for plasmid assembly. Subsequently, annotation was carried out via Prokka (version 1.14.6), a command-line tool that assists in rapid gene annotation and identifying coding sequences before data were submitted to NCBI [81]. 

Furthermore, SeqSero 2 (version 1.1.0) and SISTR (version 1.1.1) [29,30] were used for serovar prediction, while Resfinder (version 2.1) and ARG-ANNOT were used for the detection of resistance genes and chromosomal mutations [32,33]. Besides that, PlasmidFinder (version 2.0.1), PLSDB (version 2), and Plasmid Multi-locus Sequence Typing (pMLST) (version 0.1.0) were used to identify known plasmids [82,83]. *Salmonella* sequence typing was performed with PubMLST [84].

Additionally, Mobile Element Finder (version 1.0.3) was used to detect MGEs, which had inherited resistance genes [85]. Integron Finder 2.0 [86,87] and SPIFinder were also employed to detect *Salmonella* pathogenicity islands (SPI). [88]. To illustrate phylogenetic and single nucleotide polymorphism (SNPs) relationships, the CSI phylogeny (version 1.4) was used, which is available from the Center for Genomic Epidemiology website [89]. CSI phylogeny is a web-based application for detecting variation in sequencing data and building phylogenetic analyses. Following this, a phylogenetic tree was constructed using iTOL (version 6) [90].

### 4.6. Statistical Analysis

The collected data were tabulated in Excel software. Then, statistical analysis was performed using the Statistical Package for Social Sciences (SPSS) software (IBM version 20). The resistance and susceptibility percentages were also calculated. 

### 4.7. Multiple Antibiotic Resistance Index (MAR Index)

The MAR index calculation was based on previous publications [76]. MAR index = *a/b*, where *a* refers to the number of antibiotics an isolate was resistant to, and *b* represents the total number of antibiotics against which the isolates were tested.

## 5. Conclusions

This study concludes that the isolated NTS from poultry habitats exhibited significant MDR patterns. The result highlights the need to address public health concerns over the excessive usage of antimicrobials in the poultry industry. Acknowledging the limitation of environmental research centres beyond environmental studies, it is crucial for veterinary and food specialists to support a comprehensive effort and implement future collaborative action based on the One Health concept to overcome the multidisciplinary-based health issue and avoid such catastrophic outcomes. On top of that, government and local authorities should prioritise reinforcing the One Health approach with advanced molecular techniques to implement NTS monitoring and surveillance programs on antimicrobial uses as well as disease prevention in animal husbandry to prevent the spread of AMR to humans.

## 6. GenBank Accession Numbers

The *Salmonella* genomes were deposited under the BioProject genome with accession numbers PRJNA932543 and BioSamples genome with accession numbers SAMN33214859, SAMN33214860, SAMN33214861, SAMN33214862, SAMN33214863, SAMN33214864, SAMN33214865, and SAMN33214866. The Assembly genome contains the accession numbers JARJDB000000000, JARPUP000000000, JARRAP000000000, JARRAQ000000000, JARRAR000000000, JARRAS000000000, JARRAT000000000, and JARRAU000000000.

## Figures and Tables

**Figure 1 antibiotics-12-01330-f001:**
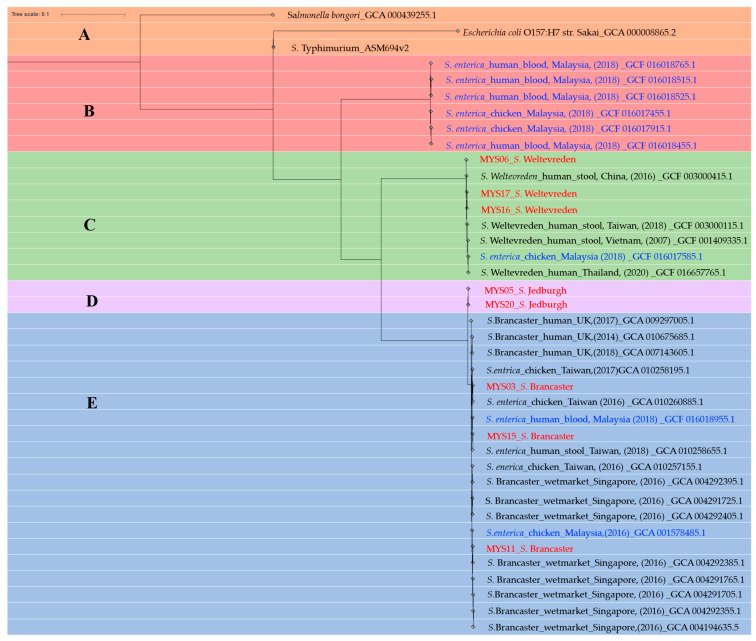
Phylogenetic tree constructed using iTOL. NOTE: The strains isolated in this study were labelled in red font and compared with isolates retrieved from the NCBI GenBank, while other strains isolated from Malaysia were labelled in blue font. The colour bands depict different groups (A–E). Group A: Form a node with references genomes; Group B: Form a node with outgroups; Group C: Form a node with S. Weltevreden serovars; Group D: form a node with S. Jedburgh serovars; Group E: form a node with S. Brancaster serovars.

**Table 1 antibiotics-12-01330-t001:** The percentage of antimicrobial susceptibility and resistance of *Salmonella* spp. isolates.

Antimicrobial Class	Antimicrobial	*Salmonella* Species			
		R/24	R (%)	I (%)	S (%)
Beta-Lactams: Penicillin	Ampicillin	15	62.5	0	37.5
	Amoxicillin/Clavulanic Acid	0	0	4.2	95.8
	Ampicillin/Sulbactam	12	50	12.5	37.5
	Piperacillin/Tazobactam	0	0	0	100
Beta-Lactams: Cephalosporins 1st Generation	Cefazolin	23	95.8	4.2	0
2nd Generation Cephalosporins	Cefuroxime	23	95.8	4.2	0
	Cefoxitin	23	95.8	4.2	0
3rd Generation Cephalosporins	Cefotaxime	0	0	0	100
	Ceftazidime	0	0	0	100
	Ceftriaxone	0	0	0	100
4th Generation Cephalosporins	Cefepime	0	0	0	100
Carbapenems	Meropenem	0	0	0	100
Monobactams	Aztreonam	0	0	0	100
Aminoglycosides	Amikacin	23	95.8	4.2	0
	Gentamicin	23	95.8	4.2	0
Fluroquinolone	Ciprofloxacin	8	33.3	16.7	50
Nitrofuran	Nitrofurantoin	1	4.2	12.5	83.3
Folate biosynthesis pathway inhibitors	Trimethoprim/Sulfamethoxazole	3	12.5	0	87.5

Abbreviations R: Resistant; I: Intermediate; S: Susceptible.

**Table 2 antibiotics-12-01330-t002:** Resistotypes of *Salmonella* spp.

Number of Antimicrobial	AMR Phenotype	*Salmonella* spp. (n = 24)	MAR Index	MDR Organism
**9**	AMP/SAM/CF/CXM/FOX/AMK/GEN/CIP/SXT	1 E	0.52	66.7%
**8**	AMP/SAM/CF/CXM/FOX/AMK/GEN/CIP	3 E, 4 S	0.47	
**7**	AMP/CF/CXM/FOX/AMK/GEN/CIP	2 S	0.42	
	AMP/SAM/CF/CXM/FOX/AMK/GEN	2 S		
	AMP/CF/CXM/FOX/AMK/GEN/SXT	2 S		
**6**	AMP/CF/CXM//FOX/AMK/GEN	1 S	0.37	
	CF/CXM/FOX/AMK/GEN/CIP	1 E		
**5**	CF/CXM/FOX/AMK/GEN	3 E, 4 S	0.32	
**1**	NIT	1 E	0.05	

Abbreviations: AMP: Ampicillin; SAM: Ampicillin/Sulbactam; CF: Cefazolin; CXM: Cefuroxime; FOX: Cefoxitin; AMK: Amikacin; GEN: Gentamicin; CIP: Ciprofloxacin; SXT: Trimethoprim/Sulfamethoxazole; NIT: Nitrofurantoin. E: Effluent sample, S: Soil sample.

**Table 3 antibiotics-12-01330-t003:** Prediction of *Salmonella* STs using PubMLST and serovars with SeqSero2.

Number of Isolates	STs and Serovar	Group	Subspecies	O Antigens	Flagellar (H) Antigens	
					Phase 1	Phase 2
*2*	*S.* Jedburgh ST 2133	E1 (O:3,10)	I	3,10	z29	-
3	*S.* Weltevreden ST 365	E1 (O:3,10)	I	3,{10},{15}	r	z6
3	*S.* Brancaster ST 2133	B (O:4)	I	1,4,12,27	z29	-

NOTE: { } = Exclusive for O factors.

**Table 4 antibiotics-12-01330-t004:** Phenotype and genotype resistance patterns, plasmid identification, pMLST, and mobile genetic elements of *Salmonella* spp. isolates.

**Code**	**Phenotypic Resistance**	**Genotypic Resistance**	**Plasmid** **and *pMLST***	**Mobile Genetic Elements**	**Chromosome** **Mutation**	**Salmonella Pathogenicity** **Islands (SPI)**
MYS03 Effluent ST 2133 *S.* Brancaster	AMP/SAM/CF/ CXM/FOX/ AMK/GEN/CIP	*aac* (*6*′)-*Iaa*, *aph*(*3*′)-*Ia* *qnrS1*, *dfrA14*, *tet*(*A*), *floR* *bla*_TEM-176_	*IncX1* *Col440I* *ColRNAI* Integron 1	Contig 18: *IncX1*- *aph (3*′)*Ia*, *bla*_TEM-176_, IS102, cn_22462_IS102. MITEEc1, Tn6024, ISEch12 *	*gyrA*: E438A # *gyrB*: A295G # *parC*: N395S # *parC*: T57S *acrB*: F28L # *acrB*: L40P #	1,3,5, 8,9
MYS05 Soil ST 2133 *S.* Jedburgh	AMP/SAM/CF/ CXM/FOX/ AMK/GEN/CIP	*aac* (*6*′)-*Iaa* *aph* (*3*′)-*Ia* *qnrS1*, *dfrA14*, *tet*(*A*), *floR* *bla*_TEM-176_	*IncX1* Integron 1	Contig 18: *IncX1-aph* (3′)*Ia, bla*_TEM-176_, IS102, cn_22462_IS102. MITEEc1, Tn6024 *	*gyrA*: E438A # *gyrB*: A295G # *parC*: N395S # *parC*: T57S *acrB*: F28L # *acrB*: L40P #	1,3,5,8, 9,12
MYS06 Effluent ST 365 *S.* Weltevreden	CF/CXM/ FOX/AMK/GEN	*aac* (*6*′)-*Iaa*	*IncFII*(*S*)- S1: A-; B-	ISEcl10, ISEam1, ISSen6, MITEEc1, ISSen1, ISSty2 *	*acrB*: F28L # *acrB*: L40P # parC: T57S	1,3,4,5,6, 9,12,13
MYS11 Soil ST 2133 *S.* Brancaster	AMP/SAM/CF/ CXM/FOX/ AMK/GEN/CIP	*aac* (*6*′)-*Iaa*, *aph* (*4)-Ia* *aph* (*3*′)-*Ia*, *aac* (*3*)-*IV* *ant* (*3*″)*-Ia*, *fosA*, *qnrS1* *sul3*, *tet*(A), *mph*(A), *lnu*(F), *bla*_TEM-1B_ *aadA17*	*IncFIA* (*HI1*)- F: A8; B-, IncHI1A- ST 16, **HCM***1_259_2* *, *IncHI1B*(*R27*), *IncN*-ST 3,12, *Col440I* Integron 1	MITEEc1, ISEch12, Tn6024, Tn5403, ISEc30, IS26 *	*gyrA*: E438A # *gyrB*: A295G # *parC*: N395S # *parC*: T57S *acrB*: F28L # *acrB*: L40P #	1,3,5, 8,9
MYS15 Soil *S.* Brancaster ST 2133	AMP/SAM/CF/ CXM/FOX/ AMK/GEN/CIP	*aac* (*6*′)-*Iaa* *aph* (*3*′)-*Ia* *qnrS1*, *dfrA14*, *tet*(*A*), *floR* *bla*_TEM-176_	*IncX1*, *Col156* *ColRNAI* Integron 1	Contig 17: *IncX1-aph*(3′)*Ia, bla_T_*_EM-176,_ IS102, cn_22462_IS102. ISEch12, Tn6024, MITEEc1 *	*gyrA*: E438A # *gyrB*: A295G # *parC*: N395S # *parC*: T57S *acrB*: F28L # *acrB*: L40P #	1,3,5, 8,9
MYS16 Soil ST 365 *S.* Weltevreden	CF/CXM/FOX/ AMK/GEN/	*aac* (*6*′)-*Iaa*	*IncFII*(*S*)- S1: A-; B-	Contig 33: *IncFII*(S)- ISEam1. ISSen6, MITEEc1, ISKpn2, ISSen1, ISEcl10 *	*acrB*: F28L # *acrB:* L40P # *parC*: T57S	3,6,9, 12,13,14
MYS17 Effluent ST 365 *S.* Weltevreden	CF/CXM/FOX/ AMK/GEN/	*aac* (*6*′)-*Iaa*	*IncFII*(*S*)- S1: A-; B-	Contig 31: *IncFII*(S)- ISEam1. ISEcl10, ISSty2, ISSen1, MITEEc1, ISSen6 *	*acrB*: F28L # *acrB*: L40P # *parC*: T57S	3,6,9, 12,13,14
MYS20 Effluent ST 2133 *S.* Jedburgh	AMP/SAM/CF/ CXM/FOX/ AMK/GEN/CIP	*aac* (*6***′**)-*Iaa* *aph* (*3***′**)-*Ia* *qnrS1*, *dfrA14*, *tet*(A), *floR*, *bla*_TEM-176_	*IncX1* Integron 1	Contig 20: *IncX1*- *aph*(*3*′)*Ia bla*_TEM-176_, IS102, cn_22462_IS102. Tn6024, MITEEc1 *	*gyrA*: E438A # *gyrB*: A295G # *parC*: N395S # *parC*: T57S *acrB*: F28L # *acrB*: L40P #	1,3,5, 8,9,12

NOTE: Amikacin: *aac(6′)-Iaa* (chromosomal cryptic gene in *salmonella*); Gentamicin: *aac(3)-IV*; Neomycin, Kanamycin: *aph(3′)-Ia*; Hygromycin: *aph(4)-Ia*; Streptomycin: *ant(3″)-Ia*; Fosfomicin: *fosA*; Ciprofloxacin: *qnrs1*; Trimethoprim: *dfrA14*; Sulfamethoxazole: *sul3*; Tetracycline: *tet*(A); Erythromycin, Azithromycin: *mph*(A); Unknown beta-lactam: *bla*_TEM-176_; Beta-lactam: Amoxicillin, Ampicillin, Piperacillin, Ticarcillin, Cephalothin: *bla*_TEM-1B_; Chloramphenicol: *floR*; Lincomycin: *lnu*(F); Streptomycin, Spectinomycin: *aadA17*. Plasmid multilocus sequence typing (pMLST). Novel allele in bold. * MEGs found in other contigs of the same isolates, # mutation not previously described.

## Data Availability

Data is publicly unavailable due to privacy and ethical restrictions. Data can be requested from the corresponding author, which will involve a few processes that include permission from the Ministry of Health, Malaysia, and the National Institute of Health, Malaysia.

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
