# Peer review of "Multidrug-Resistant Salmonella Species and Their Mobile Genetic Elements from Poultry Farm Environments in Malaysia"

_antibiotics, 2023, doi:10.3390/antibiotics12081330_

Round 1

Reviewer 1 Report

Dear authors!

Ryou need to subtract the text, insert missing words, check punctuationemarks and comments are in the attached file

Dear authors!

You need to subtract the text, insert missing words, check punctuation

Author Response

Thank you for your comments and feedback. 

Reviewer 2 Report

The manuscript by Thahir et al, titled "Multidrug-resistance Salmonella species and its mobile genetic elements from poultry farm environment in Malaysia" provides valuable information about the spread of MDR Salmonella spp. in poultry farms. I believe this communication would be of value to readers interested in antibacterial resistance. Overall, the Methods seem to be executed appropriately and the results have been presented in an understandable manner. However, I have one major comment and a few minor comments for the authors to address.

Major comment

Could the authors kindly explain which outgroups were used in the phylogenetic analysis? I think this is a critical point in the phylogeny, for example, which of the Salmonella spp. were outgroups? 

Minor comments

Line 12, italicize “Salmonella”.

Line 13, “streaked on” instead of “streaked in”.

Line 24, replace “guide” with “formulate”.

Line 26, italicize “Salmonella”.

Line 30, “Gram” instead of “gram”.

Line 33, “zoonosis” instead of “zoonotic”.

Kindly check the taxonomy of Salmonella I believe in lines 42-44, some of the species names need to be italicized, for example, it should be “Salmonella Typhimurium”. Kindly check for all species.

Line 53, “phages” instead of “phage”.

Line 61, remove “with” and “it”.

Line 111 and 113, add space between the genus and species names. Kindly also check throughout the manuscript.

Line 254, replace “in” with “on”.

Line 429, not “proceed” but “proceeded”.

Line 431, “mL” not “ml”.

Line 432 has a double space.

Needs minor editing.

Author Response

Thank you for the reviews and comments. 

changes are made in revised articles, which will be uploaded. 

Round 2

Reviewer 1 Report

The second version of the manuscript meets all the requirements and can be published

The second version of the manuscript meets all the requirements and can be published

Author Response

First of all I would like to extend my deepest appreciation for the time and effort of the editor and academic editor invested in reviewing my paper. Your expertise, meticulousness, and constructive feedback have been invaluable to me.
For the language of the paper I have consulted English editor, and changes were made accordingly.
For all the comments, I have reviewed them and amended accordingly, especially on typos.
Title was change accordingly.
Line 18 - please change MLST to sequence types- done
Tetracycline and bla genes are not correctly written- done.
Keyword has been added.
multidrug-resistance Salmonella corrected to multidrug-resistant Salmonella.
Lines 54-57: Please rephrase this sentence and specify the countries where antibiotics are still used as growth promoters/prophylaxis or add the sentence - Countries with the usage of antibiotics are listed with references.
Line 90: correction done. Previously publish studies in Malaysia
Table 1 title corrected.
Lines 120: abbreviations MAR explained.
Lines 125-126:  in silico italicized.
Lines 128-138: tools with results were rephrased.
Line 143: The aac(6')-Iaa gene -was omitted from the manuscript
Line 165: conferring resistance to the aminoglycoside group
Line 166: please rewrite the sentence - only genes for beta-lactamases WERE detected. done
Lines 151-152: the sentence was deleted.
Lines 153-158: deleted.
Lines 178-181: correction done on isolates with phenotypically resistant to fluoroquinolones. parC mutation was move to the next section 2.5.
Table 4-correction done; MIC removed.
Lines 207-215: changes were made by suggestion.
Text describing the results shown in Figure 1: (SNPs with distances less than ten indicate that they are genetically similar suggest that they share a recent common ancestor or a common source of infection, whereas those with distances greater than ten, hundreds, or thousands indicate distant origin [32]. sentences move to next paragraph which describe in general. Thank you for pointing out the error.
Line 292 : changes were made (S. Jedburgh have the same sequence type (ST 2133) with other S. Brancaster strains)
Line 446: less than 10- changes was made.
Section 4.5: version of the tools was added.